# The Mitigation Effect of Park Landscape on Thermal Environment in Shanghai City Based on Remote Sensing Retrieval Method

**DOI:** 10.3390/ijerph19052949

**Published:** 2022-03-03

**Authors:** Tian Wang, Hui Tu, Bo Min, Zuzheng Li, Xiaofang Li, Qingxiang You

**Affiliations:** 1Key Laboratory of Software Technology Research and Application of Changzhou City, Department of Computer Information and Engineering, Changzhou Institute of Technology, Changzhou 213032, China; minb45@126.com (B.M.); lixf@czu.cn (X.L.); youqx@czu.cn (Q.Y.); 2State Key Laboratory of Urban and Regional Ecology, Research Center for Eco-Environmental Sciences, Chinese Academy of Sciences, Beijing 100085, China; zuzhengli@rcees.ac.cn

**Keywords:** Urban Heat Island, park landscape, remote sensing inversion, land surface temperature, Maximum Cooling Distance and intensity, Shanghai City

## Abstract

The mitigation effects of park green space on Urban Heat Island (UHI) have been extensively documented. However, the relative effects of the configuration of park components on land surface temperature (LST) inside the park and indicators (i.e., park cooling intensity and distance) surrounding the park is largely unknown. Therefore, the main objective of this study is to explore the quantitative impacts of configuration and morphology features under different urban park scales on the cooling effect. In this study, based on Landsat-8 OLI/TIRS images on 3 August 2015 and 16 August 2020 during summer daytime, the LSTs of Shanghai City were retrieved by atmospheric correction method. Then, the relationships of park landscape features with LSTs in the park and typical indicators representing cooling efficiency of 24 parks on different grades were analyzed. The results showed that the average temperature in urban parks was, respectively, 1.46 °C and 1.66 °C lower than that in the main city of Shanghai in 2015 and 2020, suggesting that urban parks form cold islands in the city. The landscape metrics of park area (PA), park perimeter (PP), green area (GA) and water area (WA), were key characteristics that strong negatively affect the internal park LSTs. However, the park perimeter-to-area ratio (PPAR) had a significant positive power correlation with the park LSTs. Buffer zone analysis showed that LST cools down by about 0.67 °C when the distance from the park increases by 100 m. The Maximum Cooling Distance (MCD) for 2015 and 2020 had a significant correlation with PA, PC, PPAR, GA and WA, and increased sharply within the park area of 20 ha. However, the medium park group had the largest Maximum Cooling Intensity (MCI) in both periods, followed by the small park group. There could be a trade-off relationship between the MCD and MCI in urban parks, which is worth pondering to research. This study could be of great significance for planning and constructing park landscapes, alleviating Urban Heat Island effect and improving urban livability.

## 1. Introduction

According to the report from Intergovernmental Panel Climate Change (IPCC), over the past century global warming has been recognized as a profound universal problem and the increase is likely to happen faster than was predicted [1,2]. Furthermore, more than half of the world’s population lives in urban areas and this value is set to increase to 66% by 2050 [3], which would aggravate the Urban Heat Island effect (UHI). The UHI phenomenon refers to the fact that when a city grows to a certain scale, the temperature in the urban area is significantly higher than that in the non-urban areas [4], and this has been observed worldwide [5,6,7]. The increased impervious surface cover instead of evaporative vegetation surfaces and anthropogenic heat releases have been proven to be the main reasons for the UHI [8]. Overheating conditions in cites can increase urban energy consumption, raise pollution levels and may even affect the habitability of cities and potentially lead to increases in morbidity and mortality [9,10,11]. This grim situation has brought challenges to the sustainable development of cities. There are several strategies to alleviate Urban Heat Island effect in cities, such as increasing urban vegetation, using cool pavements and proper urban planning [12,13,14]. Among these cooling measures, the photosynthesis and transpiration of park green space can play a critical role in cooling and humidifying, carbon fixation and oxygen release. Urban parks, an essential component of urban green infrastructure, which not only are cold and wet islands in cities, but can influence its vicinity areas [13,15,16,17,18], are of great significance to mitigate the “Urban Heat Island” (UHI). Therefore, how to make good use of the limited urban parks and obtain the maximum ecological benefit under a high-density metropolitan area is a topic worthy of study.

Intensive studies have been conducted to assess the UHI effect for hundreds of cities around the world [7,12,13]. Land surface temperature (LST) is a crucial indicator of one component of the UHI known as the surface UHI [14,15]. Compared with in situ air temperature measurement, remote sensing provides not only the detailed information of land use/land cover, but also the LST observation with more uniform and accurate sampling [4,16]. It avoids inconsistency in data collection processes, sensor types, and other meteorological factors [17]. In recent years, the rapid development of thermal infrared remote sensing technique has greatly promoted the diversification of remote sensing inversion methods for obtaining LST, such as Linear spectral mixture analysis (LSMA) model [18,19], single channel algorithm [20], atmospheric correction or radiative transfer method [21,22,23] and split-window algorithm [24,25]. In recent years, passive microwave (PMW) satellites have developed rapidly because of their ability to penetrate clouds, although PWM data suffer from lower spatial resolution and LST retrieval accuracy compared with thermal infrared data [26]. Landsat 8 Thermal Infrared Sensor (TIRS) is the new, stable thermal infrared sensor for the Landsat project, carrying two thermal infrared bands, which provides a great benefit to the LST inversion. For example, Yu et al. (2014) [27] compared three different methods for LST retrieval from TIRS, and found that the LST inverted from the radiative transfer equation-based approach using Landsat 8 TIRS has the highest accuracy a Root Mean Square Error (RMSE) lower than 1 K. Considering that the surface UHI is more pronounced during daytime and in summer [17], this paper selects the years of 2015 and 2020 (the key period of the 13th five-year plan) to reflect the impacts of the urbanization-associated green space on urban LST at typically the same period, which has seldom been reported by other studies in Shanghai, China.

Several previous studies have found that the average LST of urban parks was 1–2 °C, and sometimes even 4–8 °C, cooler than their urban surroundings, generating a “cooling island” [7,28,29]. The spatial scope of scholars’ research about UHI mitigation effects of urban parks is generally from (1) the relationship of the urban green space landscape spatial configuration and land surface temperature (LST) [30,31,32,33], and (2) the microclimate in the parks and the impacts of their structure factors on the thermal environment of the surrounding areas [34,35,36]. Furthermore, when studying the cooling effect of the park on surrounding environment, some scholars adopted the designated buffer zone distance to calculate the average temperature around the parks and compared it with its internal temperature using remote sensing image data [28,37,38], or analyzed the impact of specific parks on the surrounding microclimate based on the field meteorological observation data [39,40,41,42]). However, the park cooling effect has mainly been characterized by a single indicator of park cooling intensity (PCI) and limited datasets (e.g., using one image or in a year), and the study of multiple parks seldom consider the impact differences of various grades of parks on the cooling effect. According to the research results of many scholars, urban parks have a significant cooling effect on urban local thermal environment with the main factors including green space, water body, impermeable layers and other park landscape composition parameters as well as park patch morphology. As we have known, the relationship between park area, shape and park cooling island is complex [43]. Zhu et al. (2021) [29] found that only parks larger than a threshold size (20 ha) would provide a larger cooling effect with the increase in park size, through a study in Jinan, China. Jaganmohan et al. (2016) [44] suggested that a number of small green spaces distributed in a city might have a stronger cooling effect than a few larger green spaces [29]. As for the shape, Cao et al. (2010) [16] quantified the cool island intensity of 92 urban parks in Nagoya, Japan, and indicated that the formation of PCI being negative affected by complex shape and fostered by shrubs and trees. In the research of Chang et al. (2007) [45], the authors thought that parks with complex shapes provided stronger cooling island effects. The above studies, with some different conclusions, suggest that the effect of configuration of park components on relative indicators (i.e., park cooling intensity and distance) is still uncertain.

Shanghai is a megacity with a subtropical monsoon climate that has undergone rapid urbanization in the last few decades. The total population has reached 24.87 million in 2021 [46], and the proportion people living of urban areas, with the highest population urbanization rate in China, reaching more than 88% [47]. Such vast urbanization has exacerbated UHI effect in Shanghai. Therefore, how to scientifically configurate the landscape elements and improve ecological service function of urban parks, especially their cooling effect, has become an urgent problem that need to be solved. Previous studies about effects of urban green space on thermal environment were reported in Shanghai, which mainly concentrated on the relationship between LST and green space pattern combined with landscape metrics from the urban landscape scale [48,49,50], the cooling effects of specific park features (e.g., park size), single landscape elements (e.g., water body) or single park type (e.g., pocket parks) [51,52,53] and thermal comfort and space use in a specific park by meteorological measurements [40]. However, a clear understanding of the composition and morphological characteristics under different scale levels of urban parks and their quantitative impact on the LST inside and in the surrounding of the parks in a time-series analyses, is still a lacking. This research can reveal the differences in the cooling capacity of park green space with different morphological characteristics over two critical years.

In view of this, based on landsat-8 OLI/TIRS remote sensing images of Shanghai in 2015 and 2020, land surface temperature was retrieved by atmospheric correction method to investigate the mitigation impacts of urban park landscapes on the thermal environment. Additionally, a total of 24 parks were selected in Shanghai City, China. The purposes of this research are to explore: (1) What kind of correlations exist between landscape composition and patch morphology of parks with LST and which are the key influencing indicators; (2) How about the Maximum Cooling Distances (MCD) and maximum cooling intensities (MCI) of parks on four different park scales? (3) Are there significant differences in heat release effects among parks of different scales and does the significance change with the development of urbanization? The results could provide some decision-making bases and references for planning and constructing Shanghai parks and alleviating Urban Heat Island effect in a hot summer, humid, subtropical large metropolis.

The paper is organized as follows. Following the description of the study area, the methodology of LST retrieval algorithm, extraction of park landscape features, buffer zone setting and data processing flow are presented in Section 2. The results, discussion and conclusions are presented in Section 3, Section 4 and Section 5, respectively.

## 2. Materials and Methods

### 2.1. Description of the Study Area

This study was conducted in Shanghai (30°40′–31°53′ N, 120°52′–122°12′ E), covering an area of 6340.5 km^2^, which is located in the central point of the north and south coast of China, where the Yangtze River and the Huangpu River meet the sea (Figure 1). It is part of the Yangtze Delta alluvial plain, with an average elevation of about 4 m and a maximum elevation of 103.4 m. According to the Köppen–Geiger climate classification, this city has a typical subtropical monsoon climate characterized by being humid and having a hot summer [54], with average annual temperature of 18.4 °C, annual precipitation of 1042.6 mm and 129 days of precipitation per year. As the famous water town in Jiangnan, Shanghai has a water area of 697 km^2^, equivalent to 11% of its total area. The zonal vegetation consists evergreen broad-leaved forest and evergreen deciduous broad-leaved mixed forest, and the non-zonal vegetation is dominated by intertidal vegetation and aquatic vegetation. Within the total area, green space covers 1242.95 km^2^, and the per capita area of park green space is 8.6 m^2^ [46]. Since the 1990s, rapid economic growth of Shanghai was accompanied by enormous urbanization in both scope and degree [55]. Currently, it is one of the four municipalities directly under the Central Government in China, belonging to the Yangtze River Delta economic circle, with a large population and developed economy and trade. It is one of China’s important foreign trade cities, known as the “magic capital”.

In the study, 24 parks in Shanghai, which have been completed and not significantly altered in the last 15 years, were selected as objects. According to Figure 1, the selected parks are mainly concentrated in the center of Shanghai City and their geometric characteristics are diverse. Location information and vector maps of the parks, water system, green spaces and road network in the study area were precisely extracted from Amap by Python crawler module. Referring to the study by Cheng X et al. (2015) [51] and taking into account reality, the parks were classified into four classes in terms of their sizes, as follows: super large (>50 ha), large (10–50 ha), medium (4–10 ha), and small (<4 ha). There are at least five parks in each grade (Table 1). The rules for demarcating the parks’ boundaries are: (1) The sidewalks and buildings in the parks were included in the park. (2) The water body within the boundaries of the parks was included in the scope of the parks, with the water body outside the boundaries excluded from the scope of the parks. (3) The traffic lines through the park were also included in the parks. This means that the vector maps of the parks were closed irregular patterns.

### 2.2. Data Sources

The remote sensing data used in this study come from two periods of Landsat-8 OLI/TIRS satellite image data, two scenes of data at about 10:24 a.m. (UTC/GMT+08:00) on 3 August 2015 (path: 118, row: 38; path: 118, row: 39) and two scenes of data at about 10:25 a.m. on 16 August 2020 (path: 118, row: 38; path: 118, row: 39), which are obtained from Geospatial Data Cloud platform of Computer Network Information Center, Chinese Academy of Sciences (http://www.gscloud.cn/search (accessed on 15 May 2021)), and imported into ENVI 5.2 software for processing to obtain two periods of land surface temperature data in Shanghai. The maximum arbitrary land cloud cover threshold adopted in this study to ensure image reliability was less than 0.28%. Landsat-8 was successfully launched by NASA on February 11, 2013. It provides global coverage every 16 days and carries two sensors—OLI Land Imager and TIRS Thermal Infrared Sensor. Landsat-8 is basically consistent with Landsat 1-7 in terms of spatial resolution and spectral characteristics. The satellite has a total of 11 bands. Band 1–7 and 9–11 have a spatial resolution of 30 m, and band 8 is a panchromatic band with a resolution of 15 m (https://earthobservatory.nasa.gov/ (accessed on 26 August 2021)).

### 2.3. Data Processing and Analysis Methods

#### 2.3.1. Land Surface Temperature Retrieval Algorithm

A previous study result has shown that the accuracy of retrieved LST based on atmospheric profile measurement is 0.6 °C, by the radiative transfer equation [56]. Therefore, this study adopted an atmospheric correction method to retrieve the land surface temperature of Shanghai in two periods [27,57]. According to the radiation transmission theory of electromagnetic wave, the thermal infrared radiation brightness value received by the satellite sensor consists of three parts: blackbody radiation brightness, atmospheric downward radiation brightness and atmospheric upward radiation brightness [27]. Its equation, which is the radiative transfer equation (RTE), is as follows:(1)$$\;L_{\lambda }=\left[\varepsilon B\left(T_{s}\right)+\left(1-\varepsilon \right)L↓\right]\tau +L↑\;$$
where $\;L_{\lambda }$ is the radiance registered in the at-sensor of the thermal band (W·m^−2^·sr^−1^·µm^−1^), B is the blackbody radiance (W·m^−2^·sr^−1^·µm^−1^), $T_{s}$ is the land surface temperature, L↓ is the downwelling path radiance, L↑ is the upwelling path radiance, τ is the atmospheric transmittance, and ε is the land surface emissivity.

The equation for *B*($T_{s}$) is as follows:(2)$$\;B\left(T_{s}\right)=\frac{\left[L_{\lambda }-L↑-\tau \left(1-\varepsilon \right)L↓\right]}{\tau \varepsilon }\;$$

Finally, $T_{s}$ can be calculated from the inversion of Planck’s law, as follows:(3)$$\;T_{s}=\frac{K_{2}}{\ln\left(\frac{K_{1}}{B\left(T_{s}\right)}+1\right)}\;$$

For the number 10 band *K*_1_ of Landsat 8, a constant of 774.89 (watt/(m^2^·srad·µm)) was applied, and for *K*_2_, a constant of 1321.08 K (Kelvin) was applied [36]. The value of ε was determined using the equation based on the NDVI threshold proposed by Sobrino et al. 2004 [56].

ENVI 5.2 software was first applied to interpret Landsat-8 OLI/TIRS images to obtain the land surface temperature of Shanghai in two periods. The experimental procedure for retrieving LST is shown in Figure 2.

For result validation, the multi-point verification method in Figure 2 represents verification of the LST values with the daily temperature data (i.e., the air temperature on 3 August 2015 and 16 August 2020 in this study) of multiple meteorological stations at the corresponding location. If the temperature trend of the retrieved results coordinates with meteorological station, the precision can be validated [50]. Hence, the corresponding temperature of five meteorological stations in Shanghai (including Pudong, Baoshan, Chongming, Hongqiao and Minhang) were used for validation (https://www.ncei.noaa.gov/maps/daily/ (accessed on 10 December 2021)). The comparison between retrieved LSTs and the data from meteorological stations are shown in Figure A1. It is clearly indicated that the trend of the curves of the retrieved LST and the air temperature from five meteorological stations were the same in 2015 and 2020. Additionally, the retrieved LST showed a larger temperature difference, which is consistent with the characteristic between the air and land surface temperature [50]. In suburban areas such as Chongming District, there are more green spaces and bodies of water with fewer buildings, resulting in its low surface temperature, which in 2015 was lower than the air temperature. After comparison and verification, the overall accuracy can meet the requirements of application.

#### 2.3.2. Temperature Division Method

In the study, natural breaks (Jenks) were taken as the temperature division method. Natural Breaks classes are based on natural groupings inherent in the data. Class breaks are identified that best group similar values and that maximize the differences between classes [58]. The features are divided into classes whose boundaries are set where there are relatively big differences in the data values. The Jenks optimization method, also called the Jenks natural breaks classification method, is a data classification method designed to determine the best arrangement of values into different classes. This is achieved by seeking to minimize each class’s average deviation from the class mean, while maximizing each class’s deviation from the means of the other groups. In other words, the method seeks to reduce the variance within classes and maximize the variance between classes. Natural breaks (Jenks), the classification method provided by ArcGIS, is the most commonly used and robust classification method. Chen J et al. (2013) [58] concluded that natural breaks (Jenks) method is of good adaptability and high accuracy on the geographical environment unit division.

#### 2.3.3. Features Extraction of Park Landscape and Buffer Zone Analysis

The cooling effect of parks on surrounding thermal environment was analyzed from the aspects of park plaque morphology and landscape composition in the study. Landscape composition represents the size of different landscape types in the park. Plaque morphology is also known as landscape shape indicator, which usually calculates the deviation between a patch shape and a circle or square with the same area to measure the complexity of its shape. The selected indicators of quantitative analysis of park landscape are shown in Table 2 [32,35,59].

Specifically, the vector maps of 24 parks, water systems, green spaces and road networks in the study area were precisely extracted from Amap by Python crawler module. Then, the basic metrics of the parks, such as the area and perimeter of parks, green spaces and water bodies were calculated by the “computational geometry” tool of the attribute table in each vector graph through ArcGIS 10.2 software. Subsequently, the indices of proportion of impermeable layers (PIL), park perimeter-to-area ratio (PPAR) and park fractal dimension (PFD), were calculated by “Field Calculator” tool based on the above attribute table, according to the formulas in Table 2.

The method of buffer zone is used to explore the cooling range and intensity of the park to its surrounding environment [29]. As the selected temperature retrieval images are from Landsat-8 OLI/TIRS with an initial spatial resolution of 100 m in the thermal infrared band, multiple graded buffers were generated outwards at this interval based on park boundaries. Some studies have reported that the range of the park cooling effect is usually around 10 s to 1000 s of meters, without exceeding 1.5 times of the width of the parks [45,60]. Therefore, 1.5 times the width of the park was set as the outermost boundary of the buffer zone in this study. Then, the cooling space was identified around each park to exclude larger areas of green space, water bodies and other influencing factors from the buffer zone, in order to homogenize the surface coverage of the analysis area as much as possible, so as to better analyze the mechanism of the park’s effect on the general urban substrate. Finally, the shapefile of each park and its buffer zone were spatially overlaid with the LST raster layers of the two phases separately to extract the average LST in the corresponding region.

The cooling effect of a park on LST can be measured in several ways [51]. Theoretically, a park’s cooling effect decreases with the distance from the edge of the park. For each park, there is an empirical relationship between the cooling intensity and the distance to the park edge. However, the cooling intensity is influenced by the surrounding environment, and may fluctuate with the distance and eventually reaches equilibrium with its surroundings; therefore, there may be several peaks on the fitness curve of the cooling intensity to the distance from the edge of the park [51]. So, we selected the cubic polynomial fitting to calculate the distance corresponding to the first peak of the cooling intensity as the MCD value [29]. In this study, two indicators were devised to quantify the cooling effect. The first is the Maximum Cooling Intensity (MCI) which was used as the indicator to quantify the cooling effect of the park according to Cheng X et al. (2015) [51]. It was calculated using equation:(4)$$\;MCI=T_{H}-T_{L}\;$$
where: $T_{H}$ = the temperature at the first peak of the cubic polynomial fit curve, $T_{L}$ = the temperature at a distance of zero from the park boundary of the cubic polynomial fit curve.

Another indicator is the Maximum Cooling Distance (MCD) which was defined as the largest distance where the Maximum Cooling Intensity (MCI) occurs. The maps of buffer analysis were produced using GIS software Arcmap 10.2 (Esri, Redlands, CA, USA).

### 2.4. Statistical Analysis

First, Pearson’s correlation analysis was utilized to assess the relationship between the mean LSTs inside and outside of the parks and each park landscape metrics. Subsequently, one way analysis of variance (ANOVA) was used to explore the significant differences between the cooling effects of different levels of parks on the surrounding thermal environment. The statistical analyses were conducted by SPSS 18.0 software. Additionally, SigmaPlot 13.0 was used as a supplementary statistical regression analysis and for mapping.

## 3. Results

### 3.1. Land Surface Temperature Features of Shanghai and the Parks in 2015 and 2020

The Landsat-8 OLI/TIRS images in 2015 and 2020 were interpreted to obtain the results of land surface temperature in Shanghai (Figure 3) by atmospheric correction method, taking advantage of natural breaks (Jenks) to divide the temperature into five grades (Table 3). From the angle of space, there was some difference in the distributions of Shanghai’s temperature between two periods of 3 August 2015 and 16 August 2020. The low-temperature zones of two periods were mainly distributed in the Yangtze River region and some areas near the East China Sea. The middle–low-temperature zones were mostly scattered in Chongming Island, the Huangpu River region in the northeast of the city as well as areas with dense water systems in the south and southeast of Shanghai. The middle–high- and high-temperature zones were mainly in the middle of Shanghai, where intensive commercial and residential districts exist. Additionally, the middle-temperature zones were primarily distributed around the surrounding zones of middle–high temperature and high temperature, and in Chongming Island and Changxing Island. From the angle of time, the overall average LST of Shanghai in 2020 (32.65 °C) was higher than that in 2015 (31.18 °C) by 1.47 °C. In addition, the highest and lowest LST in 2020 were 48.48 and 26.19 °C, respectively, 5.05 and 3.72 °C higher than that in 2015. It indicated that rapid urbanization has led to a significant increase in urban surface temperature.

In order to further compare the differences in LST and cooling capacity among these parks, the LST of the 24 parks and MCI in 2015 and 2020 are shown in Figure 4. It can be seen that there were evident differences among the LST inside the parks. The average LST and MCI of parks in 2020 were 1.73 and 0.53 °C higher than in 2015, respectively. The cooling effect of parks showed a certain fluctuation trend with the general increase in LST. For example, park 18 (Gushu park) had a relatively high LST (34.02 °C) and the strongest MCI (6.02 °C) in 2020. Park 3 (Binjiang forest park) had the lowest LST in 2015 (30.48 °C) and 2020 (31.84 °C), while its MCI scores were lower than the average temperature level. This suggested that the low LST inside the park did not imply a corresponding strong MCI. The cooling effect of the park is likely to be affected by other characteristics of the parks and the surrounding environment.

### 3.2. Correlation of Park Landscape Features with Land Surface Temperature within the Parks

To explore the quantitative relationship between the park landscape features and the mean land surface temperature within the parks, the quantitative indicators, such as park area, park perimeter, areas of different landscape elements and other metrics, were selected to analyze the correlation with the mean LST within the parks (Table 4). Additionally, the statistical regression analysis is shown as Figure 5. From the analyses, there are results as follows: (1) The park area showed a significant negative polynomial correlation with the mean LST within the parks in 2015 (*r* = −0.716, *p <* 0.01), and also with that in 2020 (*r* = −0.719, *p* < 0.01). (2) The park perimeter had a significant negative power correlation with the mean LST within the parks in 2015 (*r* = −0.690, *p* < 0.01) and 2020 (*r* = −0.677, *p* < 0.01). (3) The park perimeter-to-area ratio had a significant positive power correlation with the mean LST within the parks in 2015 (*r* = 0.632, *p* < 0.01), and also in 2020 (*r* = 0.640, *p* < 0.01). (4) The park fractal dimension exhibited an insignificant positive polynomial correlation with the mean LST within the parks in 2015 (*r* = 0.182, *R*^2^ = 0.4763) and 2020 (*r* = 0.192, *R*^2^ = 0.4636); the correlation is not significant. (5) There was a significant negative exponential correlation of the green area to the mean LST within the parks in 2015 (*r* = −0.722, *p* < 0.01) and 2020 (*r* = −0.729, *p* < 0.01). (6) The water area showed a significant negative polynomial correlation with the mean LST within the parks in 2015 (*r* = −0.498, *p* < 0.05) and with that of 2020 (*r* = −0.532, *p* < 0.05). (7) There was an insignificant negative polynomial correlation of the proportion of impermeable layers with the mean LST within the parks in 2015 (*r* = 0.312, *p* > 0.05), however, there was a significant positive polynomial correlation with that of 2020 (*r* = 0.536, *p* < 0.01).

The above correlation analysis shows that PA, PP, GA, and WA are important characteristics that negatively affect the internal LST of the park. However, the more complex the shape of the park (PPAR), the higher LST inside the parks.

### 3.3. Correlation between Park Cooling Effect and Landscape Metrics

The size and landscape composition of the public parks have an impact on the Maximum Cooling Distance and the Maximum Cooling Intensity. In the study, the buffer zones of the 24 parks were established at an interval of 100 m, with 1.5 times of the width of every park, after excluding interference space such as water bodies and impermeable layers of large size, as the area of buffer zones (Figure 6).

Two indicators of Maximum Cooling Distance (MCD) and Maximum Cooling Intensity (MCI) were devised to measure the park cooling effects on surrounding thermal environment. The Maximum Cooling Distances and the maximum cooling intensities of the 24 parks in 2015 and 2020 (Table 5) have been obtained by cubic polynomial fitting (*R*^2^ > 0.54) between cooling distances and cooling intensities for the 24 urban parks in Shanghai. In 2015, the MCD exhibited large variations from 197.30 m recorded in Shangnan Park to 1041.71 m observed in Binjiang Forest Park, with a difference of 844.41 m. While the MCI was largest in Jingnan Park and smallest in Zhongshan Park, with a difference of 2.98 °C. In 2020, the MCD was also largest in Binjiang Forest Park at 1016.19 m and smallest in Gushu Park at 201.60 m, with a difference of 814.59 m; the MCI was greatest in Gushu Park and lowest in Jing’an Park, with a difference of 5.92 °C. Notably, the Gushu Park had the smallest MCD but the strongest MCI.

Then, the correlation analysis (Table 6) of park landscape metrics with Maximum Cooling Distance and Maximum Cooling Intensity was conducted. It can be seen that the MCD for 2015 and 2020, had a significant correlation with PA, PC, PPAR, GA and WA, while the MCI of two periods had no significant correlation with any of the seven park landscape metrics. Furthermore, different from other metrics, PPAR were negatively and significantly correlated with MCD and positively correlated with the MCI both in these two years. This result implies that the more complex the shape of the park boundary, the smaller the cooling distance, but the stronger the cooling intensity. Thus, the MCD for 2015 and 2020 were fitted (Figure 7) to PA, PC, PPAR, GA and WA for further exploration.

### 3.4. One Way ANOVA of the Influence of Different Park Groups on the Cooling Effect Indicators

To further explore the impact of the park on the surrounding thermal environment, the additional study was conducted in six super large parks (>50 ha), five large parks (10–50 ha), seven medium parks (5–10 ha) and six small parks (<4 ha), in terms of the relationships between park size group and the two cooling effect indicators of Maximum Cooling Distance (MCD) and Maximum Cooling Intensity (MCI). The mean MCD of different park groups in both 2015 and 2020 decreased with park class, while there was no significant linear correlation between the mean MCI and park class in 2015 and 2020, with the medium park group having the largest cooling intensity in both two periods, followed by the small park group. It was also evident that the difference between the mean MCD of different park groups in 2015 and in 2020 was generally small, while the mean MCI of different park groups in 2020 was significantly higher than that in 2015 (Figure 8).

Then, analysis of variance (ANOVA) on the mean MCD and MCI at different park classes was further performed. According to the test of homogeneity of variances by SPSS, the significance values (Sig.) of MCD and MCI of two periods (2015 and 2020) were greater than 0.05, which meant that, the variance meets the requirement of homogeneity, after which the analysis of variance (ANOVA) could be performed. Referring to ANOVA (Table 7), the significance values (Sig.) of MCI in 2015 and 2020 were greater than 0.05, while the significance values (Sig.) of MCD were less than 0.01 in both periods. In other words, both in 2015 and 2020, the difference between the MCI of the four park groups was not significant and not statistically significant, while the opposite was true for the MCD of the parks of the four park groups, where the difference was statistically significant.

However, as the ANOVA was only able to determine whether the control variables had a significant effect on the observed variables, the next step was to conduct a multiple comparison test using the Least Significant Difference (LSD) method to further determine the exact degree of variation in MCD across the different park groups. As can be seen from the results (Figure 9), with consistency in 2015 and 2020, there was significant difference in MCD between super large and large park groups, as well as medium and small park groups, while the differences between super large and large park group and between medium and small park group were not significant. The LSD result suggested that the ability of parks of more than 10 hectares (the boundary value between large- and medium-sized parks) to affect the cooling distance was significantly enhanced.

To further assess the effects of the potential differences between the groups, the landscape metrics data were also applied to the one-way ANOVA (Figure 10). The significant difference law of PP and PPAR was similar to that of MCD, while PA and GA values represented significant differences between super large parks and other types. Among the seven landscape characteristic metrics, the differences in five metrics among four park groups were significant, except for PFD and PIL.

## 4. Discussion

### 4.1. Influence of Park Landscape Characteristics on Local Surface Temperature

As an important part of the urban landscape, the park landscape not only provides recreational areas for the surrounding residents, but also to a certain extent regulates the regional climate. The cooling effect of green spaces inside the urban park is a phenomenon that has been widely studied and validated in various regions and cities [39,42,61]. In this study, we investigated and analyzed the distribution pattern of thermal environment inside and outside the urban park through a remote sensing retrieval method. The results indicate that green spaces in parks form obvious cold islands in the city, and whether from the morphological characteristics or the composition of patches, there are key factors affecting the cooling effect. Therefore, it is important to plan the construction of urban parks in a city like Shanghai where the urban area is an expensive space. Numerous studies have shown that a park’s patchy morphology and its internal landscape metrics have a significant cooling effect on the local thermal environment [29,51]. This study found that in 2015, the average temperature of the 24 parks was 31.68 °C, 1.46 °C lower than that in the main urban area of Shanghai (33.14 °C) in that year; in 2020, the average temperature of the 24 parks was 33.42 °C, 1.66 °C lower than that in the main urban area of Shanghai (35.08 °C) in that year. This temperature difference is consistent with the conclusion drawn by Bowler et al. (2010) [62], in which the meta-analysis of data from different studies suggested that, on average, an urban park would be around 1 °C cooler than a non-green site.

This study also found that the park’s patch morphology and configuration characteristics had an impact on the thermal environment of the park. The negative relationship of mean LST of two periods to park area suggests that the LST of a park decreases with increasing park area, as well as the park perimeter. The park perimeter-to-area ratio (PPAR) had a significant positive correlation with the mean park LST (*r* = 0.63, *p* < 0.01). Additionally, the greater the PPAR, the relatively more complex the shape of the park, the easier it is to exchange material energy within the park, and the higher the mean LST of the park. However, Zhu et al. (2021) [29] found that the correlation coefficient between them was −0.39 (*p* < 0.01), indicating that parks with irregular shapes could have lower LSTs. This might be attributable to the differences in the range of PPAR of the parks and different climate backgrounds.

From the perspective of the park’s landscape composition, the negative exponential relationship of mean LST in two periods to green area suggests that the mean LST of the park decreases as the green area increases. The results of the exponential fit of the mean LST to the green area for both periods are approximately linear, and the calculation shows that for every 50-ha increase in the green space area within the park in 2015, the average park temperature decreased by approximately 0.63 °C, and for every 50-ha increase in the green space area within the park in 2020, the average park temperature decreased by approximately 0.66 °C. Therefore, when planning the internal landscape composition of urban parks, the size of the green space area should be fully considered and the green space area inside the parks should be increased as much as possible. This is because vegetation can reduce the LST through evapotranspiration and shadows, which has been proven by several studies [37,41,50].

In addition to this, the water area within the park also plays an important role in reducing the mean LST. The high specific heat capacity of the water body and the fact that evaporation from the water body can absorb some of the heat from the air results in the mean LST of the park decreasing as the water area increases [52]. The effect on temperature is relatively significant as the water area increases from 0 to 20 ha. At the same time, considering that Shanghai is a densely populated and fast-growing metropolis, its park landscape area is limited, so from the perspective of urban park landscape planning, it is more reasonable to consider the actual situation of the water body for providing greater cooling benefits. However, the results of the study show that the proportion of the impervious layer in the park had no significant effect on the mean LST of the park in 2015. Several parks selected in this study, such as Binjiang Forest Park and Jingnan Park, are surrounded by wide rivers and/or large green space (green space coverage rate exceeds 90%, shown in Table A1), which has a cooling effect on the parks and could have some impact on the results.

### 4.2. Influence of Park Landscape Characteristics on the Surrounding Thermal Environment

The scale of any cooling effect beyond the boundary of the green area is particularly important for the likely public health consequences of park greening, as park green space may not be directly accessible to all who might benefit during high temperatures [62]. Therefore, the key influencing factors and laws of the scale and intensity of cooling effect have been examined by scholars. The results reported in this paper showed that the landscape metrics of park area (PA), park perimeter (PP), park perimeter-to-area ratio (PPAR), green area (GA), water area (WA), as the critical influencing factors, influence the cooling effect of the park on the surrounding thermal environment. This result coincides with the findings of other scholars [29,37,51]. However, the effects of the park landscape features on the cooling indicator MCD were significant, while the effect on MCI was not. MCI is an indicator of temperature difference that depends not only on the LST of the park, but also on the land surface temperature around the park (Figure 3). Shanghai is located at the confluence of the Yangtze and Huangpu rivers, with a low and flat topography and a dense network of water. Many parks in Shanghai are surrounded by dense fine water flows, but in this study, buffer zones of parks did not eliminate these fine streams or some of the smaller green areas, which may have caused a slightly lower calculated surface temperature in the buffer zone than in reality.

From the plaque morphology of the park, the Maximum Cooling Distance (MCD) in 2015 (*R*^2^ = 0.70) and 2020 (*R*^2^ = 0.67) of park increased logarithmically and sharply within the park area of 20 ha but eventually reached an asymptote. After the park area exceeded the threshold (about 20–40 ha), the cooling distance tends to be gentle with the increase in the park area. Combined with the results of the MCI of the park groups, the cooling intensity of medium- (4–10 ha) and small-scale parks (<4 ha) is evidently higher than that of the super large and large group (Figure 8b). This result is in agreement with that of a previous study, in which super large parks with areas exceeding 30 ha on average were not more efficient than small parks less than 3 ha when measured by mean ratio of cooling area to park size [51]. Additionally, the Maximum Cooling Distance of the park increases linearly with increasing park perimeter. The larger the park area and the greater the park perimeter, the greater the Maximum Cooling Distance and the more significant the cooling effect of the park. In addition, the PPAR and the park’s Maximum Cooling Distance (MCD) had a negative logarithmic relationship (in 2015: *R*^2^ = 0.72, in 2020: *R*^2^ = 0.66). The smaller the PPAR, the simpler the park shape, and the more pronounced the cooling effect distance of the park on the surrounding environment. When planning and building urban parks it is necessary to take into account controlling the park PPAR from 0 to 100 to achieve a better cooling effect of park. Analyzed in terms of the park’s landscape composition, MCD of the park increased logarithmically with park green area and linearly with the park’s water area. According to the fitting results of the two periods (Figure 7), the degree of impact of the green area on the park surroundings increased sharply from 0 to 20 ha. When the vegetation area reached a certain threshold, the degree of impact tended to increase smoothly. For every 10-ha increase in the area of park water bodies, the cooling distance of park increased by 197.68 m in 2015 and 209.36 m in 2020. Transpiration from the green space and evaporation from the water body can absorb heat from the land surface and produce water vapor, which then generates wind under the action of the horizontal pressure gradient force at the land surface, resulting in a more efficient exchange of material and energy in the horizontal direction, thus mitigating the sharp rise in temperature around the park. In the planning of urban parks in metropolitan cities such as Shanghai, compared with the difficult control of water area, the proportion of green space in the parks should be maximized and the proportion of impermeable layers should be controlled while taking into account aesthetics.

### 4.3. Influence of Different Park Size Groups on the Cooling Effect Indicators

Cheng X et al., 2015, concluded that park size can explain nearly 73% of the variance in cooling distance; therefore, park size is the main factor that influences the cooling effect on land surface temperature [51]. Our results showed that the cooling distance of most parks in the study were limited within 600 m. Only a few very large parks have cooling distance over 800 m. The Maximum Cooling Distance varied significantly under different park size grades. The mean MCD values for the super large and large park groups are much larger than for medium and small ones. This is consistent with the findings of other scholars that park size does have a significant effect on the cooling effect of parks [16,37,44]. Whereas the values of mean MCD of the super large and large park groups in 2020 were larger than that in 2015, those of small and medium park group had instead shrunk slightly. With urban temperatures rising year by year, it is clear that small (<4 ha) and medium (4–10 ha) park groups have less scope for cooling influence than super large (>50 ha) and large (10–50 ha) park groups. Therefore, more large parks can be considered in large cities with a dense water network such as Shanghai, and water systems should be included in or adjacent to parks as much as possible, so that the cooling effect of parks and these water systems interact with each other to achieve a stronger cooling island effect. However, the cooling intensity of medium and small parks with less than 10 hectares should not be ignored. On the contrary, it needs to be fully utilized, especially in metropolises such as Shanghai.

### 4.4. Limitations

Land surface temperature (LST) has been widely used to describe the cooling effect of green spaces on Urban Heat Islands [16,25,55]. When using LST data rather than air temperature to study the cooling effect of parks, the intensity of cold islands is often overestimated because LST responds to direct solar radiation reaching the land surface [16]. However, remotely sensed land surface temperature data can provide more detailed spatial information and data, and they are easier to manipulate than obtaining air temperature data. Subsequent studies may consider comparing multiple inversion algorithms or using high-resolution image data combined with weather station data for calibration, etc., to make the temperature data more accurate and reliable.

According to available studies, the intensity of cold islands in the park varies during the day and night [7,45], as well as seasonally [16,63,64]. Nevertheless, only two days of Landsat-8 OLI/TIRS data during summer daytime for two periods (3 August 2015, 16 August 2020) were selected for this paper, due to the limitation of data acquisition quality and article length. The time difference between days may also induce slight errors for the comparison of LSTs between years. Although it can reflect the changes in cooling effect in different years during typical summer daytime to a certain extent, it can neither reflect the changes in cooling effect in the park by day and night, nor the changes in cooling effect by season. The amount of data should be further increased appropriately to analyze changes in the cooling effect of the park landscape. In addition, although this paper excluded larger areas of green space and water space within buffer zones, it is not precise enough to consider the effect of fine water flows and small areas of greenfield vegetation around the park on land surface temperatures. Additionally, considering that the condition of LST and configuration of greenspace may be scale dependent, a study across spatial scales should be carried out to better understand the impact mechanism of public parks on the thermal environment.

## 5. Conclusions

Understanding the effects of structure and configuration characteristics in park landscape on inside-park LST and cooling efficiency in the buffer zone is important for designing effective strategies to mitigate the amplitude of UHI. In this study, Landsat-8 OLI/TIRS images in hot-summer daytime of 2015 and 2020 representing the rapid urbanization process were interpreted, from which the LSTs were retrieved. Based on that, the relationships between park landscape features with LSTs inside the park and two cooling efficiency indicators representing change in their surrounding thermal environment were studied. We found that the average LST of urban parks was 31.68 °C in 2015 and 33.42 °C in 2020, which was 1.46 and 1.66 °C lower than that of the main urban area of Shanghai, respectively. Therefore, public parks have been performing the service functions of regulating the local thermal environment of the city. For the two indicators of MCD and MCI, the MCD results exhibited large variations ranging from 197.30 m to 1041.71 m and MCI ranges from 0.10 °C to 6.02 °C in 24 parks. The cooling distance and intensity of most parks in the study were concentrated within 600 m and 3 °C.

In terms of the park’s plaque morphology and configuration, the landscape metrics of PA, PP, GA and WA, were important characteristics that negatively affected the internal LST of the parks. However, the park PPAR had a significant positive power correlation with the park LST data. Subsequently, the MCD for 2015 and 2020 had a significant correlation with PA, PC, PPAR, GA and WA, while the MCI of two periods had no significant correlation with any of the seven park landscape metrics. Not surprisingly, larger parks had a longer cooling distance and the MCD increased logarithmically and sharply within the park area of 20 ha. However, the medium park group had the largest cooling intensity in both periods, followed by the small park group. Therefore, the cooling intensity of medium and small parks with less than 10 hectares should be fully utilized, especially in metropolises such as Shanghai with expensive space. This result also indicated that the more complex the shape of the park boundary, the smaller the cooling distance but the stronger the cooling intensity. Therefore, whether there is a trade-off relationship between the Maximum Cooling Distance and intensity of urban parks is worth pondering and continuing to research. Additionally, the original water systems should be included in or adjacent to parks as far as possible, so that the cooling effects of both can be superimposed on each other to produce a stronger cooling effect. The limitation of this paper is that the seasonal and diurnal changes in LST were not studied. At the same time, extracting the characteristics of the park combined with higher resolution images should be considered in the future. For future research we will also carry out multi-scale and multi-region comparison studies to advance our understanding of the trade-off relationship between the cooling distance and intensity of urban parks in order to maximizing the cooling effects.

## Figures and Tables

**Figure 1 ijerph-19-02949-f001:**
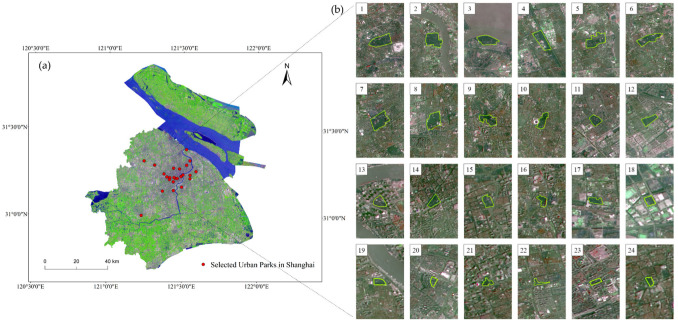
(**a**) The space distribution of selected urban parks in Shanghai. (**b**) The geometric features of the selected 24 parks and their surrounding environment. Natural true colors with RGB composition of band 4, 3 and 2, are fused with panchromatic band (band 8) to form an image base map with 15 m spatial resolution.

**Figure 2 ijerph-19-02949-f002:**
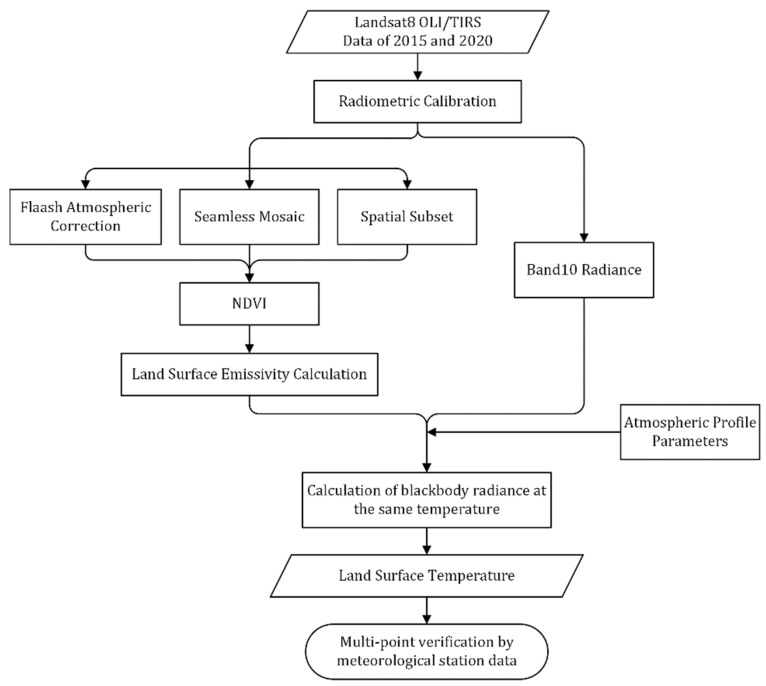
Diagram of land surface temperature retrieval process.

**Figure 3 ijerph-19-02949-f003:**
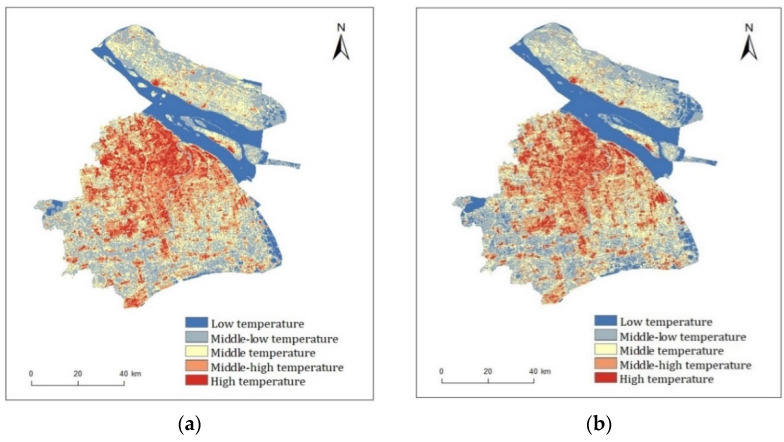
Land surface temperature derived from Landsat-8 OLI/TIRS images for the study area on 3 August 2015 (**a**) and 16 August 2020 (**b**), separately. Additionally, the temperature unit is centigrade.

**Figure 4 ijerph-19-02949-f004:**
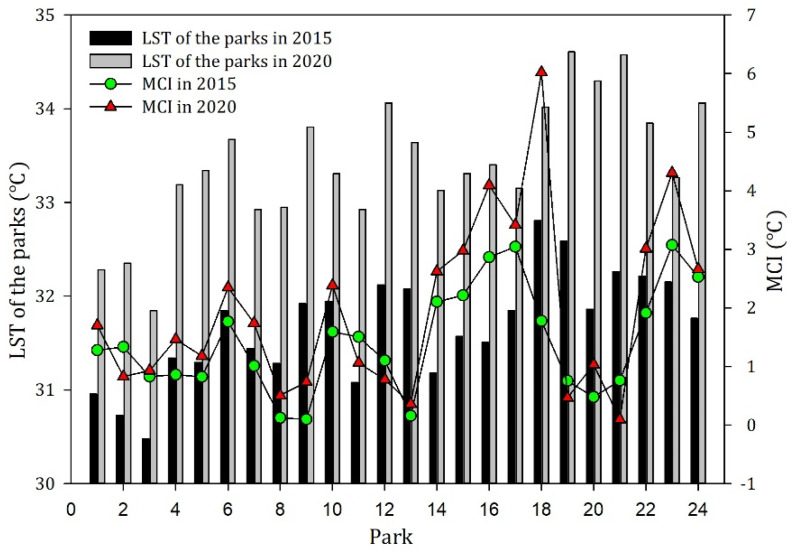
The LST inside the 24 parks and their MCIs in 2015 and 2020.

**Figure 5 ijerph-19-02949-f005:**
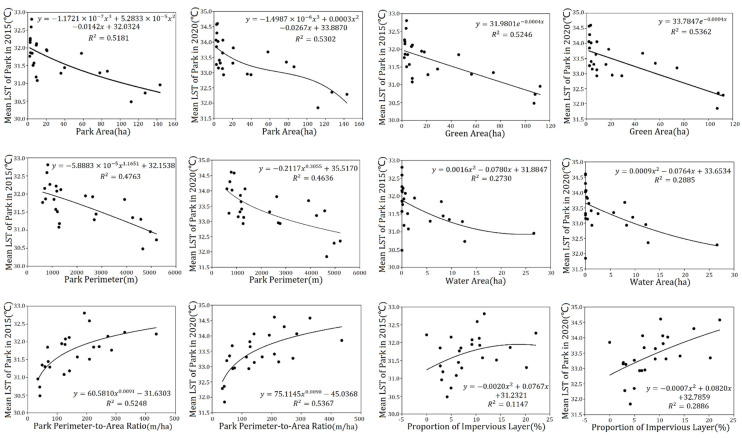
Regression analysis of park landscape metrics with mean LSTs in the parks. Note: LST = land surface temperature (the same below).

**Figure 6 ijerph-19-02949-f006:**
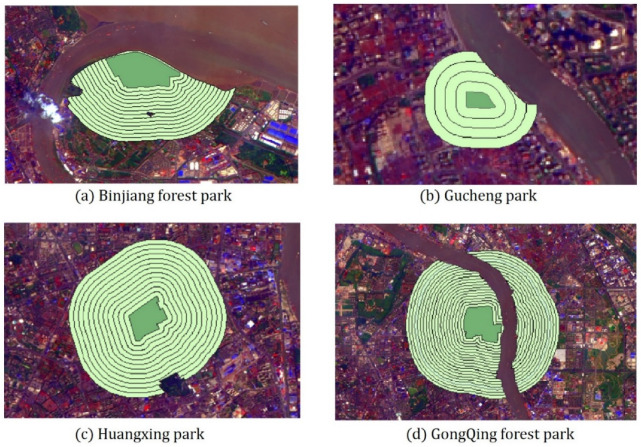
Four examples of parks’ buffer zones.

**Figure 7 ijerph-19-02949-f007:**
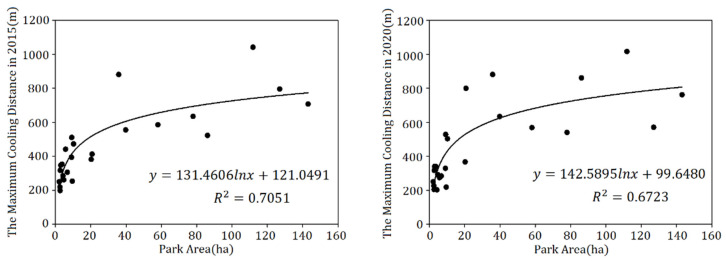
Correlation of park landscape metrics of PA, PP, PPAR, GA, WA with MCD for 2015 and 2020.

**Figure 8 ijerph-19-02949-f008:**
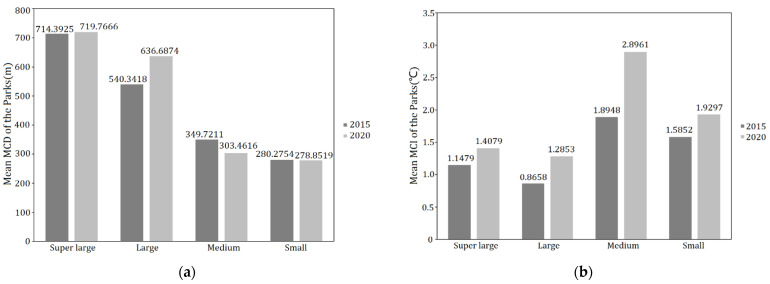
Mean MCD (**a**) and MCI (**b**) of different park groups in 2015 and 2020.

**Figure 9 ijerph-19-02949-f009:**
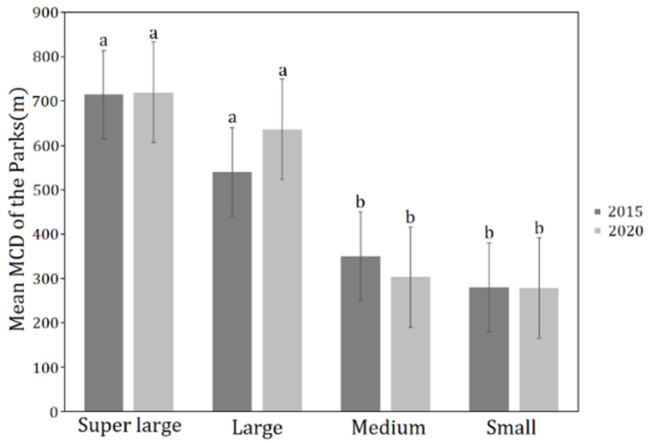
Differences in mean MCD of the parks in two periods for different park groups. Bars and line error represent mean ± standard deviation (S.D.). Lowercase letters indicate significant differences (*p* < 0.05) among park groups (the same below).

**Figure 10 ijerph-19-02949-f010:**
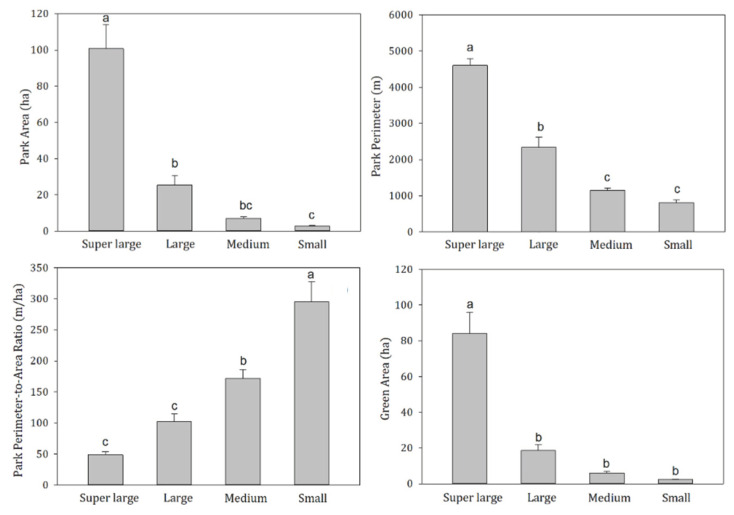
Differences in main landscape metrics (PA, PP, PPAR, GA) among different park groups.

**Table 1 ijerph-19-02949-t001:** Statistical description of 24 Urban Parks in Shanghai.

Park Group	Park Name	Park Area (ha)	Area Percentage (%)	Green Coverage (%)	Fuction	Predominant Tree Species and Main Biological Feature
Super large (>50 ha)	Century park	143.1380	0.1156	78.36	Integrated park	*Ginkgo biloba* L. (deciduous)
	GongQing forest park	127.0630	0.1026	85.08	Specialized park	*Ginkgo biloba* L.(deciduous + evergreen)
	Binjiang forest park	111.8870	0.0904	95.84	Specialized park	*Acer buergerianum* (deciduous)
	Minhang sports park	86.1472	0.0696	86.14	Integrated park	*Ginkgo biloba* L. (deciduous + evergreen)
	Shanghai botanical garden	77.9539	0.0630	72.43	Specialized park	*Cinnamomum camphora* (evergreen + deciduous)
	Daningyujinxiang park	58.0379	0.0469	79.37	Integrated park	*Ginkgo biloba* L. (deciduous + evergreen)
Large (10–50 ha)	Huangxing park	39.7864	0.0321	72.67	Integrated park	*Ginkgo biloba* L. (deciduous)
	Changfeng park	35.8347	0.0289	58.86	Integrated park	*Osmanthus fragrans* (evergreen)
	Zhongshan park	20.7742	0.0168	87.38	Integrated park	*Platycladus orientalis* (evergreen)
	Luxun park	20.2863	0.0164	78.32	Historic Garden	*Acer palmatum* (deciduous + evergreen)
	Jinqiao park	10.2522	0.0083	81.46	Community park	*Cedrus deodara* ( evergreen + deciduous)
Medium (4–10 ha)	Guyi garden	9.5335	0.0077	88.83	Historic Garden	*Salix babylonica* L. (deciduous)
	Lujiazui central green	9.2272	0.0075	83.47	Integrated park	*Magnolia denudata* (deciduous + evergreen)
	Xujiahui park	9.0939	0.0073	92.24	Integrated park	*Ginkgo biloba* L. (deciduous)
	Fuxing park	6.7610	0.0055	88.65	Historic Garden	*Platanus hispanica* (deciduous)
	Tianshan park	5.7487	0.0046	65.83	Integrated park	*Pterocarya stenoptera* (deciduous + evergreen)
	Zuibaichi park	4.7679	0.0039	96.81	Historic Garden	*Pseudolarix amabilis* (deciduous + evergreen)
	Gushu park	4.2421	0.0034	88.29	Community park	*Ginkgo biloba* L. (deciduous)
Small (<4 ha)	Gucheng park	3.7272	0.0030	89.73	Community park	*Osmanthus fragrans* (evergreen)
	Xianghe park	3.0011	0.0024	83.04	Community park	*Cinnamomum contractum* (evergreen)
	Jing’an park	2.7355	0.0022	77.84	Integrated park	*Platanus hispanica* (deciduous)
	Shangnan park	2.6522	0.0021	91.88	Community park	*Salix babylonica* L. (deciduous + evergreen)
	Jingnan park	2.5440	0.0021	95.03	Community park	*Magnolia grandiflora* (evergreen + deciduous)
	Xiangyang park	2.1324	0.0017	87.76	Community park	*Prunus serrulata* (deciduous + evergreen)

Notes: Area percentage = Percentage of the area of each park in the built-up area of Shanghai. Green coverage = Percentage of the trees or grass area in each park divided by park area. Function of each park came from the “Guiding opinions of Shanghai on the implementation of classified and hierarchical management of urban parks” jointly released by Shanghai Landscape and City Appearance Administrative Bureau and Forestry Bureau. The tree species types and main biological feature were retrieved from Baidu Encyclopedia. Because of the length of the table, the table only lists the dominant tree species in the park. The order of parks in Table 1 is consistent with that in Figure 1b.

**Table 2 ijerph-19-02949-t002:** Landscape Metrics of Shanghai’s Park and calculation method.

Classification	Landscape Metrics and Abbreviation	Calculation
Landscape composition	Green area (ha), GA	GA = green area of park
	Water area (ha), WA	WA = water area of park
	Proportion of impermeable layers (%), PIL	PIL = Ai/PA × 100%; Ai = area of impermeable layers (PA-GA-WA)
Plaque morphology	Park area (ha), PA	PA = area of park
	Park perimeter (m), PP	PP = perimeter of park
	Park perimeter-to-area ratio (%), PPAR	PPAR = PP/PA × 100%
	Park fractal dimension, PFD	D = 2 × ln(PP/4)/ln(PA) [59]

**Table 3 ijerph-19-02949-t003:** Classification of Land Surface Temperature in Shanghai.

Classification	Temperature Range in 2015 (°C)	Temperature Range in 2020 (°C)
Low temperature	<28.71	<30.03
Middle–low temperature	28.71–30.60	30.03~32.05
Middle temperature	30.60–32.25	32.05~34.06
Middle–high temperature	32.25–34.06	34.06~36.24
High temperature	>34.06	>36.24

**Table 4 ijerph-19-02949-t004:** Pearson correlation coefficients of park landscape metrics with mean LST within the parks.

Landscape Metrics	In 2015		In 2020	
	Pearson Correlation	Sig.	Pearson Correlation	Sig.
PA	−0.716 **	0.000	−0.719 **	0.000
PP	−0.690 **	0.000	−0.677 **	0.000
PPAR	0.632 **	0.001	0.640 **	0.001
PFD	0.182	0.394	0.192	0.370
GA	−0.722 **	0.000	−0.729 **	0.000
WA	−0.498 *	0.013	−0.532 **	0.007
PIL	0.312	0.138	0.536 **	0.007

Notes: * Correlation is significant at the 0.05 level (two-tailed). ** Correlation is significant at the 0.01 level (two-tailed). PA = park area, PP = park perimeter, PPAR = park perimeter-to-area ratio, PFD = park fractal dimension, GA = green area, WA = water area, PIL = proportion of impermeable layers (the same below).

**Table 5 ijerph-19-02949-t005:** Maximum Cooling Distance and Maximum Cooling Intensity of the 24 parks in Shanghai.

Park Grade	Park Name	In 2015		In 2020	
		MCD (m)	MCI (°C)	MCD (m)	MCI (°C)
Super large parks (≥50 ha)	Century park	706.9506	1.2782	762.3386	1.7007
	GongQing forest park	795.7450	1.3335	570.4900	0.8273
	Binjiang forest park	1041.7070	0.8286	1016.1858	0.9299
	Minhang sports park	522.2678	0.8592	861.2296	1.4615
	Shanghai botanical garden	634.6422	0.8223	539.7231	1.1805
	Daningyujinxiang park	585.0422	1.7656	568.6326	2.3472
Large parks (10–50 ha)	Huangxing park	554.9581	1.0104	633.7859	1.7390
	Changfeng park	881.0605	0.1235	881.0605	0.5083
	Zhongshan park	412.2149	0.0986	799.8771	0.7355
	Luxun park	381.2560	1.5903	365.7999	2.3829
	Jinqiao park	472.2197	1.5060	502.9135	1.0606
Medium parks (4–10 ha)	Guyi garden	252.9480	1.1000	217.5562	0.7764
	Lujiazui central green	510.3497	0.1572	527.7217	0.3621
	Xujiahui park	393.4313	2.1042	328.3025	2.6251
	Fuxing park	304.8691	2.2150	283.3706	2.9778
	Tianshan park	441.1852	2.8666	273.8877	4.0943
	Zuibaichi park	260.5534	3.0442	291.7883	3.4186
	Gushu park	284.7108	1.7763	201.6045	6.0184
Small parks (≤4 ha)	Gucheng park	353.0546	0.7561	338.6038	0.4656
	Xianghe park	347.0739	0.4785	339.1409	1.0331
	Jing’an park	316.6093	0.7609	316.6093	0.0988
	Shangnan park	197.3022	1.9122	203.9789	3.0141
	Jingnan park	218.6293	3.0756	225.7952	4.3050
	Xiangyang park	248.9833	2.5277	248.9833	2.6617

Notes: MCD = Maximum Cooling Distance, MCI = Maximum Cooling Intensity (the same below).

**Table 6 ijerph-19-02949-t006:** Pearson correlation coefficients of park landscape metrics with MCD and MCI.

Landscape Metrics	In 2015				In 2020			
	MCD		MCI		MCD		MCI	
	Pearson Correlation	Sig.	Pearson Correlation	Sig.	Pearson Correlation	Sig.	Pearson Correlation	Sig.
PA	0.792 **	0.000	−0.267	0.207	0.715 **	0.000	−0.292	0.166
PP	0.805 **	0.000	−0.335	0.109	0.769 **	0.000	−0.330	0.116
PPAR	−0.757 **	0.000	0.392	0.059	−0.733 **	0.000	0.330	0.115
PFD	−0.220	0.303	0.094	0.663	−0.128	0.551	0.053	0.807
GA	0.790 **	0.000	−0.254	0.232	0.715 **	0.000	−0.287	0.173
WA	0.575 **	0.003	−0.215	0.313	0.549 **	0.006	−0.203	0.341
PIL	−0.148	0.490	−0.298	0.157	−0.234	0.272	−0.194	0.365

Note: ** Correlation is significant at the 0.01 level (two-tailed).

**Table 7 ijerph-19-02949-t007:** ANOVA results of MCI and MCD among park green spaces of different scales.

		Sum of Squares	Mean Square	F	Sig.
MCI in 2015	Between Groups	3.721	1.240	1.687	0.202
Within Groups	14.705	0.735		
Total	18.426			
MCI in 2020	Between Groups	10.241	3.414	1.704	0.198
Within Groups	40.061	2.003		
Total	50.301			
MCD in 2015	Between Groups	699,250.864	233,083.621	11.130	0.000
Within Groups	418,854.943	20,942.747		
Total	1,118,105.807			
MCD in 2020	Between Groups	926,578.689	308,859.563	13.640	0.000
Within Groups	452,884.738	22,644.237		
Total	1,379,463.427

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
