# Peer review of "The Mitigation Effect of Park Landscape on Thermal Environment in Shanghai City Based on Remote Sensing Retrieval Method"

_ijerph, 2022, doi:10.3390/ijerph19052949_

Round 1

Reviewer 1 Report

The authors of this article deal with very interesting topics, such as the influence of park landscape on LST, the correlation between the park cooling effect and landscape pattern metrics, providing important information to face the urban heat island effects. Several key topics, potentially affecting the thermal environment in vegetated areas, are missing: the urban features surrounding the parks such as the urban morphology (i.e., Sky View Factor) and street orientation, the study parks’ features, such as the tree and grassland cover, tree species, leaf area density, tree height and orientation. However, further specifications and a reorganization of some sections are required, in order to clarify key aspects on the materials and methods, results, and discussion sections.

Following some major comments helpful to the authors.

  • Abstract

This section needs a substantial reorganization, especially by highlighting the aim and the phases of the research. It is also advisable to indicate the acquisition period (daytime or nighttime, and hour) by the Landsat-8 remote sensing data, the motivation, and also specify further data used.

  • Introduction

This section needs a reorganization, highlighting the key topics (such as Urban Heat Island, LST, cooling effects of vegetated areas), the atmospheric correction method, spatial tools, and statistical methods used by this study.

LST remote sensing data is an important topic for this study and requires more clarification. The following reviews on LST remote sensing data and surface UHI are suggested:   doi:10.3390/rs12162573;doi:10.3390/rs13030538;doi:10.3390/rs11010048;doi:10.1016/j.scitotenv.2020.142334).

The introduction should provide a brief presentation of the study area and the 24 parks by providing further information on green coverage (tree cover or grassland) and, if it is possible, also the predominant tree species (conifer or deciduous). In line 74 please specify the references of the “many scholars” cited: a brief explanation of the results of recent studies conducted on areas at the same latitudes as Shanghai could help the reader.

  • Materials and Methods

This section needs a reorganization and some important topics are missing, especially the climate description of the study area, the workflow of processing data, the calculation of landscape metrics, of maximum cooling distance and maximum cooling intensity indicators.

The description of the study area should be expanded by including more information on its climate classification (for example on the basis of the Köppen-Geiger climate classification, doi:10.1127/0941-2948/2010/0430). In addition, a brief description of the selected parks’ features and a map showing their geometry and location would help the reader.  

In Table 1 It would be useful to provide the percentage area value of each park compared to the overall study area. As this study strictly focuses on the mitigation effect of park landscape, information on the green coverage (i.e., the percentage of tree or grassland cover area) would be necessary. Please provide a clear explanation regarding the vector maps of the parks (in lines 128-129 the concept is not clear).

The “2.2. Data Sources” section should contain information about the acquisition period (daytime or nighttime) of remote sensing Landsat-8 data, and the cloud cover data. Please specify the reason why only two days in August 2015 and 2020 were selected in this study and discuss this aspect in the limitation study section.

In the “2.3.2. Temperature Division Method” section it is advisable to specify why natural breaks (Jenks) were used as temperature division method and to provide references for LST application studies.

In addition, please provide a better explanation of the Landscape metrics and specify how these latter were calculated. In particular, it would be advisable to extract landscape metrics data also for each park group, in order to verify potential differences between groups.

In the “2.4. Data Processing Flow” section it is necessary to explain some important topics for clarifying the study methodology framework, such as the method used to validate the LST data with the temperature data of the meteorological stations. Please specify the selected buffer area size, also providing some recent references of thermal environment studies (such as the following suggestions, doi:10.3390/rs13163154; doi: 10.1016/j.rsase.2018.04.006; doi: 10.3112/erdkunde.2021.03.03).

  • Following previous observations, the results and discussion sections require a substantial reorganization.

Author Response

Dear Editors and Reviewers:

Ms. No.: ijerph-1578646

Title: The mitigation Effect of Park Landscape on thermal environment in Shanghai City Based on Remote Sensing Inversion

We are truly grateful to your critical comments and thoughtful suggestions. Based on these comments and suggestions, we have made careful modifications on the original manuscript. All changes made to the text are marked up using the “Track Changes” function. We hope the new manuscript will meet your magazine’s standard. Below you will find our point-by-point responses to the reviewers’ comments/ questions:

Reviewer #1:

Comments and Suggestions for Authors

The authors of this article deal with very interesting topics, such as the influence of park landscape on LST, the correlation between the park cooling effect and landscape pattern metrics, providing important information to face the urban heat island effects. Several key topics, potentially affecting the thermal environment in vegetated areas, are missing: the urban features surrounding the parks such as the urban morphology (i.e., Sky View Factor) and street orientation, the study parks’ features, such as the tree and grassland cover, tree species, leaf area density, tree height and orientation. However, further specifications and a reorganization of some sections are required, in order to clarify key aspects on the materials and methods, results, and discussion sections.

Following some major comments helpful to the authors.

Responds to the reviewer’s comments: 1) Abstract This section needs a substantial reorganization, especially by highlighting the aim and the phases of the research. It is also advisable to indicate the acquisition period (daytime or nighttime, and hour) by the Landsat-8 remote sensing data, the motivation, and also specify further data used.

Response: Thanks for the Reviewer’s suggestion. We have made correction according to the Reviewer’s comments. Please see line 12-53.The time information of four remote images is also attached here:

DATE_ACQUIRED = 2015-08-03

  SCENE_CENTER_TIME = "02:24:37.7435230Z"

DATE_ACQUIRED = 2015-08-03

    SCENE_CENTER_TIME = "02:25:01.6303250Z"

DATE_ACQUIRED = 2020-08-16

    SCENE_CENTER_TIME = "02:25:01.5776640Z"

DATE_ACQUIRED = 2020-08-16

    SCENE_CENTER_TIME = "02:25:25.4729390Z"

The header file of Landsat-8 OLI/TIRS images show Greenwich Mean Time (GMT 02:24), which is 10:24 a.m. (UTC/GMT+08:00) in Beijing time.

The motivation and acquisition period, etc. are also further described in detail in the 1. Introduction, and 2. M &M, 2.2 Data Source part of the paper. Please see line 106-109 and 245-250.

Responds to the reviewer’s comments: 2) Introduction   This section needs a reorganization, highlighting the key topics (such as Urban Heat Island, LST, cooling effects of vegetated areas), the atmospheric correction method, spatial tools, and statistical methods used by this study.

Response: We have made correction according to the Reviewer’s comments. Please see line 57-195. About the atmospheric correction method, spatial tools, and statistical methods used by this study, we have made additional explanations in Section 2 Materials & methods.

Responds to the reviewer’s comments: 3) LST remote sensing data is an important topic for this study and requires more clarification. The following reviews on LST remote sensing data and surface UHI are suggested:   doi:10.3390/rs12162573;doi:10.3390/rs13030538;doi:10.3390/rs11010048;doi:10.1016/j.scitotenv.2020.142334).

Response: We have revised it according to the requirements of the Reviewer. Please see line 80-109. Thank you for your literature, which have been carefully read and added into the references.

 Responds to the reviewer’s comments: 4) The introduction should provide a brief presentation of the study area and the 24 parks by providing further information on green coverage (tree cover or grassland) and, if it is possible, also the predominant tree species (conifer or deciduous). In line 74 please specify the references of the “many scholars” cited: a brief explanation of the results of recent studies conducted on areas at the same latitudes as Shanghai could help the reader.

Response: Thanks for the Reviewer’s suggestion. We have made correction according to your comments.

1) A brief presentation of the study area Shanghai has been provided in the introduction and we added the previous studies about effects of urban green space on thermal environment reported in Shanghai. Please see line 145-179.

2) The 24 parks by providing further information on green coverage (tree cover or grassland) and, if it is possible, also the predominant tree species (conifer or deciduous) have been added in Table 1.

3)  We also added some relevant research results in Line 131-144.

Responds to the reviewer’s comments: 5) Materials and Methods

This section needs a reorganization and some important topics are missing, especially the climate description of the study area, the workflow of processing data, the calculation of landscape metrics, of maximum cooling distance and maximum cooling intensity indicators.

Response: We have revised the section according to the requirements of the reviewer.

1) The climate description of the study area has been supplied in 2.1 Description of the study area. Please see line 202-205.

2) The workflow of processing data has been added by Figure 2.

3) The calculation of landscape metrics of maximum cooling distance and maximum cooling intensity indicators has been added in Table 2 and 2.3.3 Features extraction of park landscape and buffer zone analysis.

Responds to the reviewer’s comments: 6) The description of the study area should be expanded by including more information on its climate classification (for example on the basis of the Köppen-Geiger climate classification, doi:10.1127/0941-2948/2010/0430). In addition, a brief description of the selected parks’ features and a map showing their geometry and location would help the reader. 

Response: We have made correction according to the Reviewer’s comments. Please see line 202-205, Figure 1 and Table 1.

Responds to the reviewer’s comments: 7) In Table 1 It would be useful to provide the percentage area value of each park compared to the overall study area. As this study strictly focuses on the mitigation effect of park landscape, information on the green coverage (i.e., the percentage of tree or grassland cover area) would be necessary. Please provide a clear explanation regarding the vector maps of the parks (in lines 128-129 the concept is not clear).

Response: We have provided the corresponding information of each park. Please see Table 1 and Figure 1.

Responds to the reviewer’s comments: 8) The “2.2. Data Sources” section should contain information about the acquisition period (daytime or nighttime) of remote sensing Landsat-8 data, and the cloud cover data. Please specify the reason why only two days in August 2015 and 2020 were selected in this study and discuss this aspect in the limitation study section.

Response: 1) The acquisition periods of remote sensing Landsat-8 data are all during summer daytime. Please see line 245-260. And the maximum arbitrary land cloud cover threshold adopted in this study to ensure image reliability is less than 0.28%. Please see line 252-253.

2) The reason why only two days in August 2015 and 2020 were selected in this study, which is mainly due to the limitation of article length and data acquisition.

The limitation has been discussed in the section 4.4. Please see line 722-730.

Responds to the reviewer’s comments: 9) In the “2.3.2. Temperature Division Method” section it is advisable to specify why natural breaks (Jenks) were used as temperature division method and to provide references for LST application studies.

Response: Natural breaks (Jenks), the classification method provided by ArcGIS, is the most commonly used classification method. Moreover, after natural segmentation, the classification shown in the temperature map is clearer and hierarchical. This method has a good application in the extraction of LST. This section has been further modified. Please see line 314-317. The references for LST application studies, such as; DOI 10.1007/s12665-011-1145-2; http://dx.doi.org/10.1016/j.envsoft.2016.06.021.

Responds to the reviewer’s comments: 10) In addition, please provide a better explanation of the Landscape metrics and specify how these latter were calculated. In particular, it would be advisable to extract landscape metrics data also for each park group, in order to verify potential differences between groups.

Response: We have added relevant study according to the Reviewer’s comments. Please see Table 2, Figure 10 in Section 3.4 of the Results.

Responds to the reviewer’s comments: 11) In the “2.4. Data Processing Flow” section it is necessary to explain some important topics for clarifying the study methodology framework, such as the method used to validate the LST data with the temperature data of the meteorological stations. Please specify the selected buffer area size, also providing some recent references of thermal environment studies (such as the following suggestions, doi:10.3390/rs13163154; doi: 10.1016/j.rsase.2018.04.006; doi: 10.3112/erdkunde.2021.03.03).

Response: 1) Thanks for the suggestion. We have moved the methods used to validate the LST data with the temperature data of the meteorological stations originally placed in section 2.4 to section 2.3.1 and made additions in order to make the structure clearer. Please see the line 287-302 and the validation results of Figure A1.

2) The specification of the buffer area size in this study has been provided in the section 2.3.3. Please see line 340-353.

Responds to the reviewer’s comments: 12) Following previous observations, the results and discussion sections require a substantial reorganization.

Response: We have modified according to the Reviewer’s comment. Please see the section Results and Discussion section.

We appreciate for Editors and Reviewers’ warm work earnestly, and hope that the correction will meet with approval. Thank you very much for your time and consideration.

Best regards.

Sincerely yours,

Tian Wang

2022-2-16

Reviewer 2 Report

This study investigated the cooling effects of 24 parks in Shanghai based on satellite images. Cooling effects are quantified by max cooling distance and max cooling intensity. Then the relationship between the cooling effects and landscape metrics is explored. Basically, it is worth exploring the cooling effect of green space and how to employ them to mitigate the urban heat island. However, I found this study provides nothing new to current knowledge. There are many studies, as the introduction said, focusing on the cooling effects of parks and the impact factors. This study only did another case study in Shanghai and provided nothing new in approach, results, or implications. So I suggest the authors reconsidering the research gaps and objectives again, and made something new inputs.

Some details for your reference to improve this work:

Title: what does the “remote sensing reversion” mean? It seems not common in remote sensing field.

Table1, what is the function of these park? i.e., city park, community park, or theme park, etc. Include these details in table 1.

Section 2.2., why 2015 and 2020 were used? Any specific reason to select these two years?

2.3.3, how to identify the boundary of the park? Do you have a shape file or directly recognized from the satellite images?

2.4 buffer zone, in what distance frequency?

Table 4, the LST of each park was averaged, how did the authors deal with the boundary grids? Do you only count for the grids totally within the park boundary?

Table 5, why select the cubic polynomial fitting, rather than simple linear regression?

Fig.3 three parks are near the waterbody (seems to be large river), how to exclude the impacts of the waterbody cooling, especially for d GongQing forest park, the buffer area in the other river side was also counted.  

Limitation section, Landsat-8 is for nighttime? As far as I know, Landsat is around 10am for China in daytime.

Author Response

Dear Editors and Reviewers:

Ms. No.: ijerph-1578646

Title: The mitigation Effect of Park Landscape on thermal environment in Shanghai City Based on Remote Sensing Inversion

We are truly grateful to your critical comments and thoughtful suggestions. Based on these comments and suggestions, we have made careful modifications on the original manuscript. All changes made to the text are marked up using the “Track Changes” function. We hope the new manuscript will meet your magazine’s standard. Below you will find our point-by-point responses to the reviewers’ comments/ questions:

Reviewer #2:

Comments and Suggestions for Authors

This study investigated the cooling effects of 24 parks in Shanghai based on satellite images. Cooling effects are quantified by max cooling distance and max cooling intensity. Then the relationship between the cooling effects and landscape metrics is explored. Basically, it is worth exploring the cooling effect of green space and how to employ them to mitigate the urban heat island. However, I found this study provides nothing new to current knowledge. There are many studies, as the introduction said, focusing on the cooling effects of parks and the impact factors. This study only did another case study in Shanghai and provided nothing new in approach, results, or implications. So I suggest the authors reconsidering the research gaps and objectives again, and made something new inputs.

Some details for your reference to improve this work:

Responds to the reviewer’s comments: 1) Title: what does the “remote sensing reversion” mean? It seems not common in remote sensing field.

Response: Thank for your kind suggestion. We have changed the word “reversion” to “retrieval method” in the title. Please see line 3. Other parts of the paper have also been modified accordingly.

Responds to the reviewer’s comments: 2) Table2, what is the function of these park? i.e., city park, community park, or theme park, etc. Include these details in table 1.

Response: The details of selected parks were added to Table 1 according to the Reviewer’s comments. Please see Table 1.

Responds to the reviewer’s comments: 3) Section 2.2., why 2015 and 2020 were used? Any specific reason to select these two years?

Response: The land use has changed dramatically in various types for the last 10 year in Shanghai. Given that the 12th Five-Year Plan ranged from 2011 to 2015 and the 13th Five-Year Plan ranged from 2016 to 2020, thus 2015 and 2020 were selected in the study to analyze the LST change and the urban park benefits for the urban ecology services function, especially the cooling function with the development of urbanization. The study can reflect the impacts of the urbanization-associated green space on urban LST at typically the same period. Please see lines 106-109 for this description in the paper.

Responds to the reviewer’s comments: 4) 2.3.3, how to identify the boundary of the park? Do you have a shape file or directly recognized from the satellite images?

Response: The shapefiles of the parks were precisely extracted from Amap by Python crawler module in this study. Please see line 229-230.

Responds to the reviewer’s comments: 5) 2.4 buffer zone, in what distance frequency?

Response: In the study, the buffer zones of the 24 parks were established at an interval of 100 meters, with 1.5 times of the width of every park. The detailed reason description is in section 2.3.3. Please see line 340-353.

Responds to the reviewer’s comments: 6) Table 4, the LST of each park was averaged, how did the authors deal with the boundary grids? Do you only count for the grids totally within the park boundary?

Response: Firstly, the shapefiles of the parks were precisely extracted from Amap by Python crawler module in this study. Secondly, in ArcGIS, the vector range was used to extract the mean value in the corresponding LST raster, thus not only count for the grids totally within the park boundary. Similar methods can be seen in the calculation of average value LST in buffer zones. Please see line 351-353. And the rules for demarcating the parks’ boundaries are in line 230-239.

Responds to the reviewer’s comments: 7) Table 5, why select the cubic polynomial fitting, rather than simple linear regression?

Response: Theoretically, a park’s cooling effect decreases with the distance from the edge of the park. For each park, there is an empirical relationship between the cooling intensity and the distance to the park edge. But the cooling intensity is influenced by the surrounding environment, and may fluctuate with the distance and eventually reaches equilibrium with its surroundings; therefore, there may be several peaks on the fitness curve of the cooling intensity to the distance from the edge of the park. So, we selected the cubic polynomial fitting to calculate the distance corresponding to the first peak of the cooling intensity as the maximum cooling distance. Please see the line 354-371.

Responds to the reviewer’s comments: 8) Fig.3 three parks are near the waterbody (seems to be large river), how to exclude the impacts of the waterbody cooling, especially for  GongQing forest park, the buffer area in the other river side was also counted.

Response: The large body of water adjacent to the parks, as the interference area, which were removed from the study area. As the buffer area in the other river side is also part of GongQing forest park, the buffer area in the other river side was counted for ensuring the integrity of the park study.

Responds to the reviewer’s comments: 9) Limitation section, Landsat-8 is for nighttime? As far as I know, Landsat is around 10 am for China in daytime.

Response: After inquiry, the MTL.txt header file of Landsat-8 OLI/TIRS images show Greenwich Mean Time (GMT 02:24), which is 10:24 a.m. (UTC/GMT+08:00) in Beijing time (during the hot-summer daytime). Thanks for your reminding and we also have made corresponding correction in Limitation section.

We appreciate for Editors and Reviewers’ warm work earnestly, and hope that the correction will meet with approval. Thank you very much for your time and consideration.

Best regards.

Sincerely yours,

Tian Wang

2022-2-16

Reviewer 3 Report

Thank you for sharing your research. In general, the paper is good-written and properly supported by references. The problem is clearly introduced. The method is clearly presented. Results and conclusions are logical and clear.  Graphical information is purposefully presented. I have only a few minor remarks:

  • The aim of the study should be clearly introduced in the Abstract.
  • I would recommend to include a brief paragraph at the end of the Introduction in order to indicate the structure of the document. This helps the reader to have an accurate idea about the organization and facilitates the reading.
  • Line 363, line 366, line 447. “the writers”? Please avoid “we”, “our”, etc.
  • The Conclusions section should be expanded with the limitations of the study and future lines of research.

Author Response

Dear Editors and Reviewers:

Ms. No.: ijerph-1578646

Title: The mitigation Effect of Park Landscape on thermal environment in Shanghai City Based on Remote Sensing Inversion

We are truly grateful to your critical comments and thoughtful suggestions. Based on these comments and suggestions, we have made careful modifications on the original manuscript. All changes made to the text are marked up using the “Track Changes” function. We hope the new manuscript will meet your magazine’s standard. Below you will find our point-by-point responses to the reviewers’ comments/ questions:

Reviewer 3#:

Comments and Suggestions for Authors

Thank you for sharing your research. In general, the paper is good-written and properly supported by references. The problem is clearly introduced. The method is clearly presented. Results and conclusions are logical and clear. Graphical information is purposefully presented. I have only a few minor remarks:

Responds to the reviewer’s comments: 1) The aim of the study should be clearly introduced in the Abstract.

Response: Thanks for the Reviewer’s comment. We have added the aim of the study in the Abstract. Please see line 18-20.

Responds to the reviewer’s comments: 2) I would recommend to include a brief paragraph at the end of the Introduction in order to indicate the structure of the document. This helps the reader to have an accurate idea about the organization and facilitates the reading.

Response: Thanks for the Reviewer’s suggestion. The paragraph has been inserted at the end of the Introduction. Please see line 191-195.

Responds to the reviewer’s comments: 3) Line 363, line 366, line 447. “the writers”? Please avoid “we”, “our”, etc.

Response: We have made correction according to the Reviewer’s comments. Please see line 583, 591.

Responds to the reviewer’s comments: 4) The Conclusions section should be expanded with the limitations of the study and future lines of research.

Response: The Conclusions section has been expanded according to the Reviewer’s comments. Please see Section 5 line 781-786.

 We appreciate for Editors and Reviewers’ warm work earnestly, and hope that the correction will meet with approval. Thank you very much for your time and consideration.

Best regards.

Sincerely yours,

Tian Wang

2022-2-16

Round 2

Reviewer 1 Report

  1.  It would be advisable to explain how information on the percentage green coverage area, and the type of tree species was obtained. Please specify the references or the data, methods, and tools used.
  2. Please briefly explain the “Multi-point verification by meteorological station data” (mentioned in Figure 2) in the material and method section.

  3. Data from Table 2 regarding “Calculation” and “Description” of landscape metrics is repetitive and does not add relevant information about the methodological approach. Instead, It would be preferable to include in the text which tool and data were used for landscape metric calculation (for example, if GIS tools were used and remote sensing data were applied).

Author Response

Dear Editors and Reviewers:

Ms. No.: ijerph-1578646

Title: The mitigation Effect of Park Landscape on thermal environment in Shanghai City Based on Remote Sensing Inversion

Thank you very much for your letter and valuable suggestions on our manuscript. Based on these comments, we have made careful modifications on the previous manuscript and checked the full text. All changes made to the text are marked up using the “Track Changes” function. We hope that the revision will meet your magazine’s standard and look forward to hearing from you soon. Below you will find our point-by-point responses to the reviewers’ comments/ questions:

Comments and Suggestions for Authors

Reviewer #1:

Responds to the reviewer’s comments: 1) It would be advisable to explain how information on the percentage green coverage area, and the type of tree species was obtained. Please specify the references or the data, methods, and tools used.

Response: (1) Firstly, the vector range of parks and green spaces of 24 parks were precisely extracted from Amap by Python crawler module. Then, the basic metrics of the parks, such as the area and perimeter of parks, green spaces and water bodies were calculated by the "computational geometry" tool of the attribute table in each vector graph through ArcGIS 10.2 software. Finally, the green space coverage of each park in table 1 was calculated by the trees or grass area in each park divided by the park area.

The calculation method is described in the notes below table 1. And we have also added details in the corresponding places in the text. Please see line 180-182 and 192-193 and 284-293.

(2) The tree species types are retrieved on Baidu Encyclopedia by entering the park name. The note has been also added below table 1. Please see line 196.

(3) In addition, the functions of the parks have been carefully checked and revised based on the "Guiding opinions of Shanghai on the implementation of classified and hierarchical management of urban parks" officially released by Shanghai landscape & City Appearance Administrative Bureau and Forestry Bureau from August 19, 2020.

https://lhsr.sh.gov.cn/ysqzzdgk/20201110/70260bf7decb41fab7621dd37d664f70.html

The note has been also added below table 1. Please see line 194-195.

Responds to the reviewer’s comments: 2) Please briefly explain the “Multi-point verification by meteorological station data” (mentioned in Figure 2) in the material and method section.

Response: The multi-point verification method in Figure 2 represents verification of the LST values with the daily temperature data (i.e., the air temperature on August 3, 2015 and August 16, 2020 in this study) of multiple meteorological stations at the corresponding location. In addition, the meteorological station data in this paper was downloaded from the wheata system, a big data system of agrometeorology, and the data source was mainly from the National Oceanic and Atmospheric Administration (NOAA). Relevant website: https://www.ncei.noaa.gov/maps/daily/

We have briefly explained in the text. Please see line 242-251.

Responds to the reviewer’s comments: 3) Data from Table 2 regarding “Calculation” and “Description” of landscape metrics is repetitive and does not add relevant information about the methodological approach. Instead, It would be preferable to include in the text which tool and data were used for landscape metric calculation (for example, if GIS tools were used and remote sensing data were applied).

Response: Thank you for your suggestion. The duplicate column “Description” of landscape metrics in Table 2 has been deleted. And the calculation process and tools of the landscape metrics were added specifically in the text. Please see line 284-293 and table 2.

We appreciate for Editors and Reviewers’ warm work earnestly, and hope that the correction will meet with approval. Once again, thank you very much for your comments and suggestions.

Best regards.

Sincerely yours,

Tian Wang

2022-2-27